# Reduction in Placental Metal and Metalloid in Preeclampsia: A Case–Control Study

**DOI:** 10.3390/nu16060769

**Published:** 2024-03-07

**Authors:** Yanhui Hao, Wen Yu, Jiaying Wu, Yingyu Yue, Yanting Wu, Hefeng Huang, Weibin Wu

**Affiliations:** 1Obstetrics and Gynecology Hospital, Fudan University, Shanghai 200011, China; yhao13@fudan.edu.cn (Y.H.);; 2The International Peace Maternity and Child Health Hospital, School of Medicine, Shanghai Jiao Tong University, Shanghai 200030, China; 3Shanghai Medical College, Fudan University, Shanghai 200032, China; 22301050201@m.fudan.edu.cn

**Keywords:** case–control, preeclampsia, pregnancy hypertension, metal and metalloid, placenta, selenium, essential elements

## Abstract

Preeclampsia is a primary placental disorder, with impaired placental vascularization leading to uteroplacental hypoperfusion. We aimed to investigate differences in metal and metalloid content between the placentas of women with preeclampsia and healthy controls. This was a case–control study in 63 women with preeclampsia and 113 healthy women. Clinical data were obtained from medical records. Inductively coupled plasma mass spectrometry (ICP-MS) was used to measure the placental metals and metalloids content. Compared with healthy control subjects, preeclampsia was associated with a significantly lower concentration of essential elements (magnesium, calcium, iron, copper, zinc, and selenium) in the placental tissue. After multivariable adjustment, an interquartile range (IQR) increase in selenium concentration was associated with a reduced risk of preeclampsia with an OR of 0.50 (95% CI: 0.33–0.77). The joint effects of multiple selected metals and metalloids were associated with a reduced risk of preeclampsia. The lower placental magnesium, chromium, iron, zinc, and selenium concentrations of preeclampsia cases indicate a potential link to its pathogenesis. It also provides an intriguing avenue for future research in revealing the underlying mechanisms and potential intervention strategies for preeclampsia.

## 1. Introduction

Preeclampsia (PE) is a serious condition that can have detrimental effects on both the mother and fetus during pregnancy. It is characterized by hypertension and proteinuria or other signs of end organ damage after 20 weeks of gestation, and it is considered a pregnancy-specific disorder caused by the placenta and cured only by delivery [1,2]. Preeclampsia is associated with an elevated risk of adverse maternal and neonatal outcomes, including maternal eclampsia, preterm birth, small for gestational age (SGA), fetal growth restriction, and perinatal mortality [3,4]. One proposed mechanism is the overproduction of reactive oxygen species (ROS) by an ischemic or stressed placenta, which may lead to endothelial cell dysfunction, hypertension, and clinical manifestations of preeclampsia [5]. There are several risk factors associated with an increased likelihood of developing preeclampsia. These include pre-pregnancy obesity and nutrient deficiencies [6,7,8,9]. Certain metals and metalloids, such as magnesium (Mg), calcium (Ca), iron (Fe), copper (Cu), zinc (Zn), and selenium (Se), play important roles in various biochemical reactions and metabolic pathways [10]. One of their key functions is to serve as a cofactor, aiding enzymes in catalyzing chemical reactions. During pregnancy, these metals and metalloids are particularly crucial for fetal growth and development [11,12].

Ca is an important physiological element that is abundant in the human body and acts as a cofactor for numerous enzymes, facilitating their catalytic processes [13]. Mg is an essential element in the human body that plays a crucial role in numerous enzymatic reactions [14]. Mg supplementation has been shown to be beneficial in preventing seizures associated with preeclampsia [15,16]. Additionally, Mg is also a Ca antagonist; both Ca and Mg play important roles in maintaining blood pressure [17,18,19]. Fe requirement increases dramatically during pregnancy due to the expansion of maternal plasma and blood volumes, as well as the growth and development of the fetus. Cells with high metabolic rates require more iron and are at greater risk of dysfunction during Fe deficiency [20].

During pregnancy, essential elements such as chromium (Cr), manganese (Mn), cobalt (Co), copper (Cu), zinc (Zn), and selenium (Se) are essential for various physiological processes [21]. For example, matrix metalloproteinases (MMPs) are a family of zinc- and calcium-dependent endopeptidases involved in cytotrophoblast migration and invasion of the uterine wall and in the remodeling of the spiral arteries [22]. Se is an important component of antioxidant selenoproteins and plays a critical role in regulating antioxidant status [23]. Poor Se status has been associated with an increased incidence of preeclampsia [24]. Mohammad Safiqul et al. found considerably lower circulating levels of Zn, Cu, Mn, and Fe in serum of preeclampsia patients in comparison to the control group. Imbalances or deficiencies in these elements may disrupt element homeostasis and contribute to the development of the condition [25]. Furthermore, exposure to certain pollutant metallic and metalloid elements, such as cadmium (Cd), lead (Pb), and arsenic (As), may increase the risk of preeclampsia. These elements can induce oxidative stress, endothelial dysfunction, and immune abnormalities [26,27,28].

A recent meta-analysis has suggested that certain essential elements, e.g., Ca, Se, Zn, and Mn, are found at lower levels in serum of women with preeclampsia [29]. However, due to the accumulation effect of metals and metalloids, only evaluating the concentration of metals and metalloids in the blood cannot reflect the actual exposure level in placental tissue during pregnancy. Given that severe preeclampsia is characterized by placental hypoperfusion and ischemia, it is crucial to examine the distribution of metals and metalloids in placental tissue and their potential for bioaccumulation. Understanding these factors is essential for comprehending the mechanisms of intrauterine exposure associated with preeclampsia [30,31].

Thus, this study aims to determine the distribution of metals and metalloids in preeclampsia and healthy control placentas and investigate the association between placental metals and metalloids exposure and preeclampsia. This research could provide valuable information on the role of metals and metalloids in the development of preeclampsia and potentially help identify preventive strategies or interventions.

## 2. Materials and Methods

### 2.1. Study Design and Study Participants

This study had a case–control design and included 63 women with preeclampsia and 113 healthy control women with similar sociodemographic characteristics and an equivalent distribution of offspring genders between the two groups. All participants were Shanghai residents of Han ethnicity with singleton pregnancies. The placenta samples were collected at the biobank of the International Peace Maternity and Child Health Hospital (IPMCH) Shanghai, China, from 2016 to 2018. Exclusion criteria for participation in this study included multiple pregnancies, conceived by in vitro fertilization, recorded pre-existing hypertension, or pre-existing chronic hypertension or heart disease, diabetes, cancer, or renal failure before pregnancy.

We reviewed outpatient charts for blood pressure (systolic blood pressure (SBP) ≥140 mmHg or diastolic blood pressure (DBP) ≥90 mmHg on two clinically measured occasions) and development of proteinuria after 20 weeks of gestation, and we additionally reviewed inpatient hospital charts for women who had a diagnosis or discharge code indicating preeclampsia. Preeclampsia was defined according to the International Society for the Study of Hypertension in Pregnancy, and hypertensive disorders of pregnancy (HDP) included gestational hypertension, preeclampsia, chronic hypertension (essential or secondary), or preeclampsia superimposed on chronic hypertension. Women diagnosed with chronic hypertension before pregnancy were excluded. Healthy control subjects were women with no pregnancy complications or adverse birth outcomes, matched to the preeclampsia group for maternal educational level, gravidity, parity, and mode of delivery. The sociodemographic characteristics, medical history, and birth outcomes of neonates were collected from the hospital information system. We calculated maternal pre-pregnancy body mass index (BMI) as weight in kilograms divided by height in meters squared. Maternal BMIs were categorized into three groups recommended by the Working Group on Obesity in China, and a BMI cutoff of 24.0 kg/m^2^ was used to define overweight [32].

This study was approved by the Institutional Review Board (IRB) of the IPMCH affiliated with the School of Medicine, Shanghai Jiao Tong University (GKLW2023-006 and date of approval 7 February 2023). Eligible women signed informed consent forms prior to donating their biospecimens to the biobank of IPMCH. Further informed consents were waived in this study as approved by the IRB.

### 2.2. Assay of Metal and Metalloid in Placenta

Placentas were collected promptly following delivery or termination of pregnancy, assigned codes, frozen, and stored in −80 °C freezers at the biobank of IPMCH until analysis. Subsequently, all placental samples were sent in a batch to the assessment laboratory at Fudan University for determination using inductively coupled plasma mass spectrometry (ICP-MS, NexION 300X, PerkinElmer, Norwalk, MA, USA) [28]. The operators were blind to the groups of samples. Before measurement, placentas were defrosted at 4 °C and rinsed with deionized water to remove residual blood, and subsequently dried using filter paper. Approximately 0.5 g of wet weight was weighed on an analytical balance and placed in a polytetrafluoroethylene digestion vessel; then, 5 mL of concentrated nitric acid (70% ultrapure nitric acid from BASF, Ludwigshafen, Germany) was added. Microwave digestion (CEM MARS, Charlotte, NC, USA) was performed until complete dissolution, followed by dilution with deionized water for analysis (the detailed parameters are shown in the Appendix A). The standard solutions used for the analysis were purchased from PerkinElmer (standard solution, PerkinElmer mixed standard 3, 10 μg/mL, 5% nitric acid).

### 2.3. Statistical Analysis

Continuous variables that followed a normal distribution were reported as the mean (±standard deviation) and compared by using an independent samples *t*-test. For variables that did not follow a normal distribution, the median (interquartile range, IQR) was reported and compared using the Wilcoxon–Mann–Whitney test. The chi-square test was used for categorical variables. Correlations between the quantified metal and metalloid were analyzed with Spearman’s rank correlation (α = 0.05).

Associations between metal (or metalloid) exposure and the odds of preeclampsia diagnosis were examined by using one-pollutant models. To assess confounding, we began with an unadjusted model and then added the confounders (adjusted model) defined as covariates known to be associated with exposure and outcomes but not in the causal pathway. Confounding factors included maternal age, education level (high school or below, college or above), parity (primiparous, multiparous), pre-pregnancy BMI (categorized into 3 groups: underweight (<18.5), normal weight (18.5–23.9), overweight and obesity (≥24)), and gestational age at delivery. We calculated ORs and 95% CIs with logistic regression to estimate increases in odds of preeclampsia per IQR increase in placental metal (or metalloid) concentration.

Considering that both insufficient and excessive levels of metal (or metalloid) in the body can lead to adverse health effects and multiple metals (or metalloids) coexisting simultaneously, we employed a Bayesian Kernel Machine Regression (BKMR) model to examine the overall correlation between measured metal (or metalloid) exposure and preeclampsia risk. The BKMR is a nonparametric statistical approach for estimating the joint health effects of multiple concurrent exposures [33,34]. We evaluated the joint effects of metals and metalloids on preeclampsia by using BKMR with a probit link function. This model was adjusted for the same set of confounders used in the logistic analysis, including maternal age, pre-pregnancy BMI, and gestational age at delivery.

The BKMR analyses and plotting were conducted by using the “bkmr”, “ggplot2”, and “corrplot” packages in R software (version 3.5.2). The remaining statistical analyses were conducted in SAS 9.4.

## 3. Results

### 3.1. Characteristics of the Study Participants

In total, 63 women with preeclampsia and 113 healthy women were included in the final analysis (Figure 1). Demographic characteristics for all participants are shown in Table 1. Women in the preeclampsia group were more likely to be overweight or obese in the pre-pregnancy stage (30.2% vs. 12.4%, *p* = 0.01) compared with healthy controls who delivered full-term without pregnancy complications. And the fetal outcomes were worse among this group, including 14 infants (22.2%) that were born with fetal growth restriction (FGR), 28 (44.4%) that had low birth weight, and 30 (47.6%) that experienced premature birth at less than 37 weeks of gestation. Newborns of mothers with preeclampsia also had a shorter gestational age, averaging at 36.0 ± 3.0 weeks, compared to newborns in the healthy control group, with a gestational age of 39.2 ± 0.9 weeks.

### 3.2. Metal and Metalloid in Placenta

Placental concentrations of Mg, Ca, Cr, Mn, Fe, Cu, Zn, Se, and As in the preeclampsia group were significantly lower than those in the healthy group (Table 2). Of all the minerals detected, Ca, Mg, Fe, and Zn were the most abundant in the placental tissue. The median concentration of Ca in placental tissues was highest in both the preeclampsia group (3341.57 μg/L) and the healthy control group (2355.98 μg/L). In addition to Ca, Mg demonstrated the second highest median concentration in both the preeclampsia group (1809.58 μg/L) and the healthy control group (1457.59 μg/L).

A further comparison was conducted within the preeclampsia group. The data showed that in the subgroup of preeclampsia with FGR, there were lower levels of placental essential elements. Specifically, placental Fe showed a significantly lower concentration in the FGR-complicated subgroup (1055.93 μg/L vs. 806.10 μg/L, *p* < 0.001) (Table 3).

According to gestational age, the preeclampsia group was divided into three subgroups: an extremely preterm group (delivery before 28 weeks), a preterm group (gestational age >28 weeks but <37 weeks), and term preeclampsia. In the three subgroups, no consistent pattern was observed among the levels of placental essential elements. For example, the placental Mg was lowest in the preeclampsia without the preterm delivery group, while placental Fe showed a significant gradual increase from the extremely preterm group (826.86 μg/L) to the preterm group (923.98 μg/L) and finally to the full-term group (1064.32 μg/L), with a significant difference among the subgroups (*p* = 0.032) (Table 4).

We also compared the concentrations of placental essential elements between the healthy control group and the preeclampsia subgroup. Except for Co and Cd, there were significant between-group differences in the concentrations of other types of essential elements (Appendix A).

A notable inconsistency in the correlation between the placental Mg concentration and gestational weeks is illustrated in the healthy control and preeclampsia groups (Figure 2). Among the three groups of healthy control individuals, categorized based on pre-pregnancy BMI as normal weight, overweight/obese, or underweight, placental Mg exhibits an increasing trend with gestational weeks across all weight groups. However, in the preeclampsia group, there was no significant increase in Mg concentration as gestational weeks progressed.

In scatter plots of other metals (or metalloids) with gestational weeks, similar distributions and trends were observed. Specifically, in the healthy control group with normal pre-pregnancy weight, we observed a gradual “cumulative” increase in metal (or metalloid) concentrations with increasing gestational weeks. However, in the preeclampsia group, the concentration of metals (or metalloids) in the placenta did not show a significant increase with increasing gestational age. It is important to note that within the group of healthy participants, there were variations in the relationships between minerals and gestational weeks when they were divided into subgroups based on the mother’s pre-pregnancy BMI (Appendix A). Therefore, in the subsequent multivariate logistic analysis, along with maternal age, we included gestational age and pre-pregnancy BMI as covariates.

Spearman correlation coefficient matrices for placental metal and metalloid concentrations are given in Appendix A. Significant positive correlations were observed between multiple metals (or metalloids) in the placental tissues of both healthy controls and the preeclampsia group. Cu-Se (0.88), Mg-Se (0.88), Mn-Cr (0.85), and As-Zn (0.82) exhibited strong correlations. In both groups, the correlation between Co and other elements was the weakest.

### 3.3. Placental Metals and Metalloids and Preeclampsia Risk

In Table 5, we show unadjusted and multivariable-adjusted associations of metals and metalloids with preeclampsia risk. In the unadjusted models, all detected metals and metalloids except for Cd were significantly associated with preeclampsia risk. Even after adjusting for maternal age, pre-pregnancy BMI, and gestational age at delivery, the associations between preeclampsia risk and the five elements (Mg, Cr, Fe, Zn, and Se) remained statistically significant. For example, increment of placental Se (1.38 μg/L) was associated with a lower risk of preeclampsia in both unadjusted and adjusted models (OR: 0.78, 95% CI: 0.70–0.86; OR: 0.50, 95% CI: 0.33–0.77, respectively). Marginal associations with reduced risk of preeclampsia were also observed in placental Mg and Fe (OR = 0.99; 95% CI: 0.99–1.00).

Joint effects of the metals and metalloids on preeclampsia by using the BKMR model indicate that an increase in measured metal and metalloid concentrations corresponded to a decreased preeclampsia risk (Figure 3).

## 4. Discussion

Using a hospital-based case–control study, we measured and compared the levels of metals and metalloids in placental tissues from women with preeclampsia and healthy control subjects in the central area of Shanghai, China. The results showed that eight physiological elements, namely, Mg, Ca, Cr, Mn, Fe, Cu, Zn, and Se, were significantly lower in preeclampsia placentas. After adjusting for potential covariates, this study found an inverse association between five of the placental metals and metalloids (Mg, Cr, Fe, Zn, and Se) and preeclampsia risk. Thus, inadequate estimated metal and metalloid content in placenta may be associated with the occurrence of preeclampsia.

The traditional placental etiology hypothesis suggests that poor remodeling of the spiral arteries of the uterus and placenta is associated with early onset preeclampsia and several other major obstetric syndromes, including fetal growth restriction, placental abruption, and spontaneous premature rupture of membranes [35]. The processes of trophoblast fusion, invasion, and the remodeling of the spiral arteries all require extracellular matrix (ECM) degradation and the participation of certain metal ions as ligands for key enzymes, such as Zn and Ca for the MMP family [36,37]. Therefore, a deficiency in these ions in the placenta of preeclamptic cases may impact trophoblast cell function and contribute to the development of preeclampsia. Our data have identified significantly lower amounts of placental essential elements in preeclampsia cases. This finding suggests that monitoring and maintaining appropriate electrolyte levels are essential for ensuring optimal maternal health and a healthy pregnancy.

Our study revealed a positive correlation between the concentration of metals and metalloids in the placenta and the duration of pregnancy among healthy control subjects. However, in cases of preeclampsia, there was no significant increase in placental metal or metalloid concentration as gestational age advanced. Moreover, upon conducting additional subgroup analysis, significant disparities were identified between cases of preeclampsia and the healthy control group. Notably, cases of preeclampsia with fetal growth restriction exhibited an even lower metal and metalloid content, particularly in terms of Fe. These results align with the placental pathology observed in preeclampsia patients [36]. On the other hand, previous studies have demonstrated that metals like Ca and Fe undergo a significant transfer to the developing fetus, with the highest concentration occurring in the third trimester weeks [38,39]. However, due to severe complications, preeclampsia commonly leads to premature termination of pregnancy. Consequently, the placental tissue in these cases may not have initiated high-concentration metal and metalloid transportation. This further leads to a reduction in metal and metalloid content in the placental tissue of preeclampsia cases. Nevertheless, the precise underlying biological mechanism of preeclampsia remains to be elucidated. Based on this case–control study, we were unable to establish a clear causal relationship between mineral exposure and the risk of developing preeclampsia. Further prospective research, incorporating multiple time points and diverse sample types, is imperative to gain deeper insights into this matter.

Previous investigations into the impact of metal exposure and its association with preeclampsia primarily relied on human biomonitoring samples, specifically monitoring metal concentrations in maternal blood or urine [6,7,25,40,41]. For example, an analysis conducted by Liu et al. utilizing the Boston Birth Cohort found that Mn levels in the red blood cells of pregnant women had a protective effect against the development of preeclampsia; however, no protective effect was found for Se [6]. Another case–control study, conducted in Bangladesh, found significantly lower serum levels of Fe and Zn in preeclampsia patients compared to healthy pregnant women [25]. Similarly, a retrospective cohort study involving 2186 women in Guangdong, China, reported that elevated levels of blood Mg during mid-term pregnancy were associated with a decreased risk of developing preeclampsia [40]. Furthermore, a recent meta-analysis comparing Se levels in women with preeclampsia and normotensive controls found a significant correlation between low Se levels and preeclampsia [42]. In our present study, we also discovered a positive correlation between Mg, Fe, Zn, and Se levels in placental tissue and a reduced risk of preeclampsia, which is consistent with previous studies.

In a study conducted by Aleksandar et al., examining the distribution of trace elements in placental tissue among 105 healthy Caucasian women, the median concentrations of trace elements were reported as follows: Cu at 858 ng/g (approximately 0.85 μg/L), Se at 140 ng/g (approximately 0.14 μg/L), Mn at 91.3 ng/g (approximately 0.09 μg/L), and Cr at 9.69 ng/g (approximately 0.01 μg/L) [43]. Another study, which focused on Cr accumulation or burden in 50 healthy women who were residents of Michigan, reported that Cr was found in all human placenta samples, ranging from 0.02 to 1.2 ppm (μg/L) [44]. However, our present study revealed a different order of these four trace elements in placental tissue within the healthy control group, specifically Cr, Cu, Se, and Mn. Furthermore, the metal levels observed in our study were comparatively higher. In addition, as the placental Cr was higher in the healthy control group compared to the preeclampsia case group, the placental Cr appears to have a “protective” effect against the risk of developing preeclampsia. Nonetheless, research investigating the association between preeclampsia and placental metals remains limited. The divergent findings across studies may be attributed to the differences in specimens, dietary habits, environmental factors, and genetic susceptibilities among different study populations.

The placenta plays a pivotal role as an interface for the exchange of maternal physiological and environmental signals with the developing fetus. As such, it holds significant potential in assessing prenatal exposures within the framework of developmental origins of health and disease (DOHaD) [45,46]. It is crucial to acknowledge that the accurate assessment of maternal absorbed metals and metalloids necessitates considering not only their concentration in maternal blood and retention within the placenta but also the quantity transferred to the fetus. By simultaneously collecting samples of maternal blood, placental tissue, and umbilical cord blood during delivery, we can enhance our understanding of preeclampsia and facilitate the development of more targeted approaches for both its prevention and treatment, addressing the associated conditions more effectively.

This study does have certain limitations that should be acknowledged. Firstly, we were unable to differentiate between various speciation and oxidation states of metal ions within the placental tissue samples. These variations are crucial factors that can impact the physiological effects and toxicity of these metals [10]. Secondly, our analysis primarily focused on the examination of placental tissue, which may limit our understanding of the comprehensive distribution of metals and metalloids among the mother, offspring, and placenta. Lastly, the relatively small sample size during the follow-up period may introduce potential selection bias, highlighting the need for further replication in larger cohort studies to validate the findings obtained from this study.

## 5. Conclusions

Our study revealed notable variations in the median concentrations of metals and metalloids within placental tissue between the preeclampsia and healthy control groups. The lower median concentrations of Mg, Cr, Fe, Zn, and Se in the placental tissue may be associated with the development of preeclampsia. However, it is important to recognize that the pathophysiology of preeclampsia is multifaceted, and further investigation is necessary to explore the underlying mechanisms and the impact of essential elements imbalances on the pathophysiology of preeclampsia.

## Figures and Tables

**Figure 1 nutrients-16-00769-f001:**
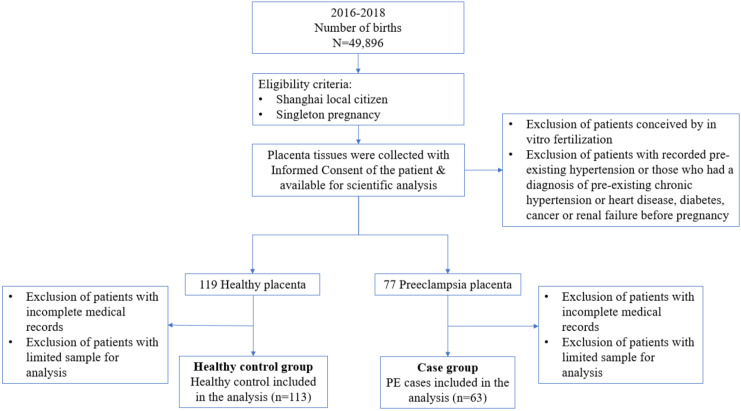
Participant flow chart.

**Figure 2 nutrients-16-00769-f002:**
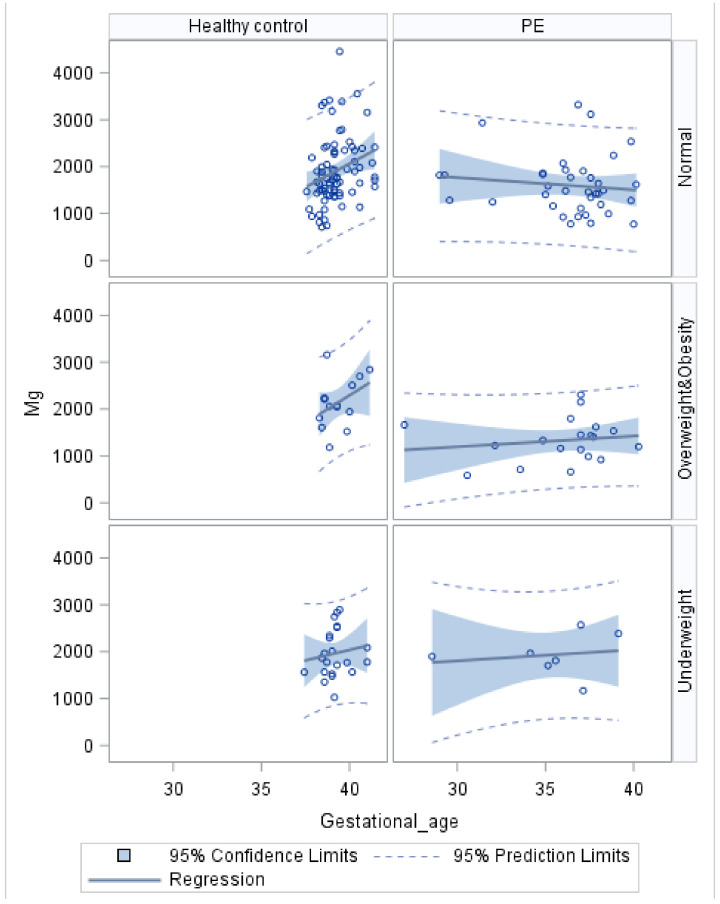
Placental Mg distribution across gestational age (weeks) in the healthy control group and the preeclampsia (PE) group, categorized by maternal pre-pregnancy body mass index (BMI) groups. BMI as weight in kilograms divided by height in meters squared. Maternal BMI were categorized into three groups recommended by the Working Group on Obesity in China: BMI < 18.5 was defined as Underweight, BMI ≥ 24.0 kg/m^2^ was defined as overweigh & Obesity.

**Figure 3 nutrients-16-00769-f003:**
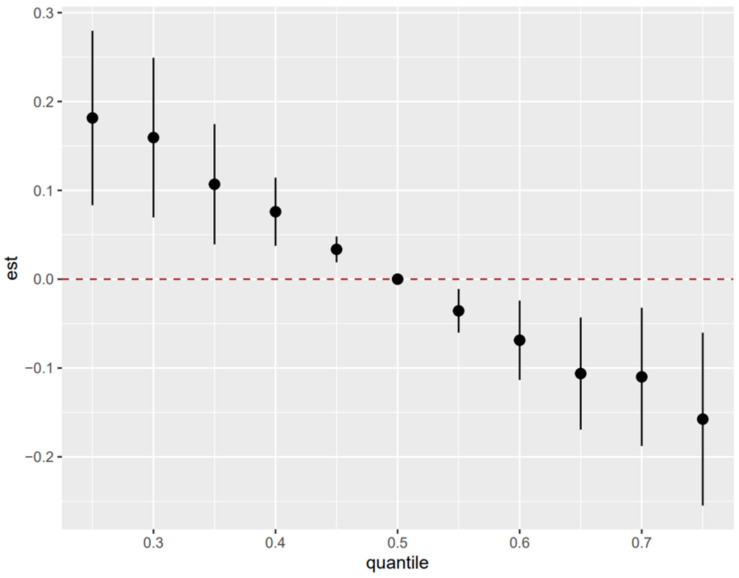
The potential effects of metal and metalloid mixture on the occurrence of preeclampsia. All minerals at particular percentiles were compared to all the chemicals at their 50th percentile. Est can be interpreted as it indicates that an increase in measured metal and metalloid concentrations corresponded to a decreased preeclampsia risk. The model was adjusted for maternal gestational age, maternal age, and pre-pregnancy BMI.

**Table 1 nutrients-16-00769-t001:** Descriptive characteristics of mothers and children in healthy control and preeclampsia group.

	Healthy Controls (*n* = 113)	Preeclampsia (*n* = 63)	*p*-Value
Maternal age, mean ± SD	30.5 ± 3.5	31.7 ± 4.4	0.061
*n* (%)			
20–24 y	5 (4.4%)	2 (3.2)	0.218
25–29 y	47 (41.6)	24 (38.1)	
30–34 y	49 (43.4)	23 (36.5)	
≥35 y	12 (10.6)	14 (22.2)	
Prepregnancy BMI Category ^1^, *n* (%)			**0.01**
<18.5	22 (19.5)	7 (11.1)	
18.5–23.9	77 (68.1)	37 (58.7)	
≥24	14 (12.4)	19 (30.2)	
**Educational level**, *n* (%)			0.949
High school and lower	40 (35.4)	22 (34.9)	
College and higher	73 (64.6)	41 (65.1)	
Gravidity, *n* (%)			0.438
1	60 (53.1)	31 (49.2)	
≥2	53 (46.9)	32 (50.8)	
**Parity, *n* (%)**			0.939
Nulliparous	92 (81.4)	51 (80.9)	
Multiparous	21 (18.6)	12 (19.1)	
**Mode of delivery, *n* (%)**			0.152
Vaginal birth	91 (80.5)	56 (88.9)	
Cesarean section	22 (19.5)	7 (11.1)	
**Fetal Growth Restriction, *n* (%)**			**<0.001**
YES	0	14 (22.2)	
NO	113 (100)	49 (77.8)	
**Gestational Diabetes Mellitus, *n* (%)**			**<0.001**
YES	0	13 (20.6)	
NO	113 (100)	50 (79.4)	
**Child Characteristics**			
Boy, *n* (%)	44(38.9)	24(38.1)	0.912
**Gestational age, mean** **± SD, wk**	39.2 ± 0.9	36.0 ± 3.0	**<0.001**
**Preterm birth, *n* (%)**	0	30 (47.6)	**<0.001**
**Birth weight category, *n* (%)**			**<0.001**
<2500 g	0	28 (44.4)	
2500–3999 g	106 (93.8)	32 (50.8)	
≥4000 g	7 (6.2)	3 (4.8)	

^1^ BMI, body mass index (calculated as weight in kilograms divided by height in meters squared).

**Table 2 nutrients-16-00769-t002:** Median (IQR) concentrations (μg/L) of placental metals and metalloids measured in healthy control and preeclampsia group.

	Healthy Controls (*n* = 113)	Preeclampsia (*n* = 63)	*p*-Value
Mg	1809.58 (1500.01, 2340.72)	1457.59 (1163.36,1833.15)	**<0.001**
Ca	3341.57 (2208.47, 6907.35)	2355.98 (1249.65,3142.29)	**<0.001**
Cr	18.65 (10.65, 22.65)	9.47 (7.25,15.02)	**<0.001**
Mn	2.51 (1.7, 3.09)	1.89 (1.25,2.51)	**<0.001**
Fe	1554.66 (1239.73, 1960.29)	966.88 (827.89,1377.96)	**<0.001**
Co	4.83 (0.13, 6.98)	4.26 (2.49,6.18)	0.376
Cu	13.69 (11.29, 17.04)	11.33 (9.86,13.51)	**<0.001**
Zn	165.31 (141.18, 199.22)	118.1 (97.75,139.53)	**<0.001**
Se	3.21 (2.56, 4.15)	2.46 (2.08,2.84)	**<0.001**
Cd	0.24 (0.17,0.36)	0.21 (0.15,2.35)	0.397
As	0.34 (0.29,0.39)	0.24 (0.19,0.28)	**<0.001**

Values are presented as median (P25, P75), *p* value was based on the Wilcoxon–Mann–Whitney test. Abbreviations: As: arsenic; Ca: calcium; Cd: cadmium; Co: cobalt; Cr: chromium; Cu: copper; Fe: iron; Mg: magnesium; Mn: manganese; Se: selenium; Zn: zinc.

**Table 3 nutrients-16-00769-t003:** Median concentrations (μg/L) of metal and metalloid subgroups by fetal growth restriction (FGR) in preeclampsia group (*n* = 63).

	Preeclampsia without FGR * *n* = 49	Preeclampsia with FGR * *n* = 14	*p*-Value
Mg	1496.13 (1199.6, 1897.06)	1278.40 (923.33, 1703.87)	0.079
Ca	2419.17 (1399.99, 3369.36)	1648.74 (762.83, 2363.44)	0.055
Cr	9.59 (7.51, 16.37)	8.42 (6.74, 11.79)	0.209
Mn	1.95 (1.35, 2.51)	1.82 (1.15, 2.43)	0.608
Fe	**1055.93 (883.22, 1435.87)**	**806.10 (647.66, 894.21)**	**<0.001**
Co	4.26 (2.28, 6.69)	4.21 (3.11, 5.71)	0.869
Cu	11.37 (10, 13.3)	10.77 (8.42, 13.54)	0.418
Zn	118.39 (98.69, 142.76)	109.96 (76.45, 128.68)	0.150
Se	2.47 (2.18, 2.84)	2.25 (1.59, 2.74)	0.203
Cd	0.22 (0.15, 2.35)	0.20 (0.17, 1.38)	0.647
As	0.23 (0.19, 0.29)	0.25 (0.19, 0.28)	0.504

* Values are presented as median (P25, P75), *p* values was based on the Wilcoxon–Mann–Whitney test for the comparison between subgroups of preeclampsia with or without FGR. Abbreviations: As: arsenic; Ca: calcium; Cd: cadmium; Co: cobalt; Cr: chromium; Cu: copper; Fe: iron; Mg: magnesium; Mn: manganese; Se: selenium; Zn: zinc.

**Table 4 nutrients-16-00769-t004:** Median concentrations (μg/L) of metal and metalloid subgroups by gestational age in preeclampsia group (*n* = 63).

	Term Preeclampsia *n* = 33	Preterm Preeclampsia * *n* = 20	Extremely Preterm Preeclampsia * *n* = 10	*p*-Value
Mg	1437.01 (1166.51, 1757.01)	1645.19 (1160.97, 1846.6)	1472.48 (1226.08, 1820.85)	0.928
Ca	2457.94 (1687.26, 3598.42)	2006.48 (762.83, 2692.06)	2335.17 (883.41, 2990.36)	0.159
Cr	8.66 (7.22, 13.45)	9.41 (7.32, 17.79)	12.6 (8.46, 19.08)	0.399
Mn	1.89 (1.16, 2.31)	1.91 (1.2, 2.69)	2.36 (1.86, 2.69)	0.342
Fe	**1064.32 (894.21, 1453.42)**	**923.98 (789.85, 1130.09)**	**826.86 (706.09, 1006.56)**	**0.032**
Co	4.44 (2.28, 6.69)	3.44 (1.39, 5.26)	5.12 (2.93, 5.76)	0.388
Cu	10.82 (9.53, 13.51)	11.48 (10.3, 13.53)	10.15 (9.88, 12.74)	0.699
Zn	115.98 (92.23, 136.54)	126.45 (98.94, 163.39)	123.07 (106.05, 138.17)	0.350
Se	2.46 (2.08, 2.89)	2.45 (2.13, 2.82)	2.48 (1.9, 2.74)	0.895
Cd	0.20 (0.14, 6.94)	0.21 (0.19, 0.95)	0.27 (0.17, 0.28)	0.839
As	0.22 (0.19, 0.27)	0.26 (0.2, 0.3)	0.28 (0.26, 0.31)	0.075

* Preterm, delivery after 28 weeks of gestation and less than 37 weeks of gestation; extremely preterm, delivery before 28 weeks of gestation. Values are presented as median (P25, P75), *p* values were based on the Wilcoxon–Mann–Whitney test for the comparison between subgroups of preeclampsia with or without extremely preterm. Abbreviations: As: arsenic; Ca: calcium; Cd: cadmium; Co: cobalt; Cr: chromium; Cu: copper; Fe: iron; Mg: magnesium; Mn: manganese; Se: selenium; Zn: zinc.

**Table 5 nutrients-16-00769-t005:** Odds ratio and 95% CI for preeclampsia in relation to a quartile increase in each metal or metalloid exposure.

	Model 1 *			Model 2 ^†^		
	OR (95% CI)		*p*-Value	Adjusted OR (95% CI)		*p*-Value
Mg	0.99	(0.99, 1.00)	<0.001	**0.99**	**(0.99, 1.00)**	**<0.001**
Ca	0.94	(0.90, 0.98)	0.006	0.99	(0.96,1.03)	0.760
Cr	0.95	(0.93, 0.97)	<0.001	**0.90**	**(0.85, 0.95)**	**<0.001**
Mn	0.75	(0.66, 0.86)	<0.001	0.83	(0.59,1.17)	0.278
Fe	0.99	(0.99, 1.00)	<0.001	**0.99**	**(0.99, 1.00)**	**<0.001**
Co	0.93	(0.87, 0.98)	0.013	0.99	(0.88, 1.11)	0.821
Cu	0.95	(0.93, 0.97)	<0.001	0.92	(0.84, 1.01)	0.088
Zn	0.99	(0.99, 1.00)	<0.001	**0.98**	**(0.97, 0.99)**	**<0.001**
Se	0.78	(0.70, 0.86)	<0.001	**0.50**	**(0.33, 0.77)**	**0.002**
Cd	0.98	(0.89, 1.08)	0.661	1.08	(0.95, 1.23)	0.222
As	0.26	(0.10, 0.68)	0.006	0.66	(0.31, 1.40)	0.276

* Model 1 was an unadjusted model. ^†^ Model 2 was adjusted for maternal age, pre-pregnancy BMI, and gestational age at delivery. Abbreviations: As: arsenic; Ca: calcium; Cd: cadmium; Co: cobalt; Cr: chromium; Cu: copper; Fe: iron; Mg: magnesium; Mn: manganese; Se: selenium; Zn: zinc.

## Data Availability

Data is contained within the article and Appendix A.

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
