# Peer review of "Reduction in Placental Metal and Metalloid in Preeclampsia: A Case–Control Study"

_nutrients, 2024, doi:10.3390/nu16060769_

Round 1
Reviewer 1 Report
Comments and Suggestions for Authors
Preeclampsia is a disease of theories. The authors realised a case-control study demonstrated the association of a significantly lower concentration of some essential elements in the placental tissue in these cases.
The topic is essential if we refer to pregnancy with preeclampsia, and the association can be a real one, but it needs more studies. The results can contribute to finding a mechanism for this pathology and discovering potential preeclampsia intervention strategies.
The study identified the concentration of more essential elements and evidenced the relationship between the values of each element and preeclampsia. Larger cohort studies need to validate the findings obtained in this study. The figures and the tables are well-designed and comprehensible.
The conclusions are consistent, and they address the main question.
I have some observations:
- In the abstract, more abbreviations of the essential elements are used with no explanations.
- In Materials and Methods, please clearly write the inclusion criteria.
- In line 124, please change the comma to a dot.
- At line 117, please correct 4 â—¦C.
- In figure 1 is recommended to write the total number of patients in the studied period. One of the inclusion criteria is that the patients give birth. After exclusions from the number of patients, patients must remain without placentas because they remain patients. The abbreviation PE – needs an explanation.
- At line 312 is written Liu et al. and reference 6 is Zhang, Y. Line 324 refers to a study of Aleksandar et al., and refers to reference 11 which is de Baaij. Please verify all the references.
- The references are appropriate. Articles from the references from 31 to 41 are not cited in the article.
Reviewer 2 Report
Comments and Suggestions for Authors
Dear authors
congratulations
very nice manuscript, very interesting.
every effort to implement knowledge regarding PE
must be supported
the manuscrip is nice
the study well conducted
results well presented
discussion well described
i would suggest some minor revisions
1) please add a table with a summary of findings to help reader to better understand the results
2) please add few senteces regarding the high risk associated to PE such as of eclamspia wich impact on maternal and fetal morbidity and morality (PMID: 35317697
PMID: 32980358
best regards
Reviewer 3 Report
Comments and Suggestions for Authors
The authors studied placental tissues obtained from patients with Preeclampsia (PE) and they demonstrated a lower placental concentration of Mg, Cr, Fe, Zn and Se.
There are several comments that may improve the manuscript:
1. An Histopathological examination of the placentas can provide functional information about reduced trophoblastic invasion of the spiral arteries, as suggested by the authors.
2. In Table 1 the gestational age of women with PE , between 33 and 39 weeks, shows the absence of cases of early preeclampia es 26-28 weeks. Placentas obtained from an earlier gestational age may yield more significant results.
3. It would be helpful to verify a possible correlation between placental concentration of metals and metalloids and fetal growth curves
4. References must be updated
Comments on the Quality of English Languagegood
Reviewer 4 Report
Comments and Suggestions for Authors
Introduction
Line 48 - the sentence in citation number 15 refers to a review-type publication that does not explicitly mention the importance of elemental Ca in the subject under discussion in its title - so I propose here to add the original source citation of the publication indicating the potential of Ca in blood pressure maintenance.
2. Materials and Methods
2.2. Assay of metal and metalloid in placenta
In the reviewer's opinion, while this section quite clearly, succinctly, and neatly describes the procedure for the preparation of the sample of the placenta and its further procedure for the downstream analytic approach, what is missing here is a brief description of the methodology of the sample analysis itself, for what parameters the analyses were finally conducted, at what injections of samples (were there any their concentrations/dilutions?), time of measurement, etc. It is worth supplementing this part of the manuscript with this information at this point.
3. Results
3.1. Characteristics of the study participants
Line 157-163 - In the text indicating specific numerical data (percentages), there is a missing reference to where these values come from (Table no. 1).
Shouldn't the description of Table number 1 look like this?: "Table 1. Descriptive characteristics [n (mean) +/- SD (% of n)] of mothers and children in control and preeclampsia group.", or something like that. This is the impression you get after recording the data in the table. This is a suggestion to consider.
Line 192-200 - Do the authors have data on the quantitative variability of the analyzed elements between the previously cited groups: an extremely preterm group, a preterm group, and preeclampsia at term precisely, in relation to control women? Because it is not included in tables 4 or S1 and S2. What would the statistical significance of the results look like if they (numeric values) were assessed against an appropriately divided control group, as the authors subsequently pointed out: pre-pregnancy BMI as normal weight, overweight/obese, or underweight? (However, another form of presentation of discrepancies was presented in the form of Figures S1-S10). I’m asking out of curiosity.
Discussion
Line 323 - At the end of the sentence, there is a lack of appropriate citations of selected original publications.
In this chapter, I would suggest replacing the term "research" with "study". Suggestion to consider.
Noteworthy in the discussion is the concise, transparent way of trying to translate the observed changes in the particular elements analyzed into possible pathological consequences in the placenta.
As rightly emphasized, further prospective research, incorporating multiple time points and diverse sample types, is imperative to gain deeper insights into this matter.It is right that the study limitations were highlighted. It was also good that the possibility/necessity of simultaneously collecting samples of maternal blood, placental tissue, and umbilical cord blood during delivery was pointed out to better understand PE development pathogenesis and its potential prevention. The Authors have correctly highlighted the need to increase the size of the analyzed groups, in particular with PE, divided i.e. into previously mentioned groups.
Round 2
Reviewer 3 Report
Comments and Suggestions for Authors
The manuscript is suitable for publication in its present form.